# One and Two-Step In Vitro-In Vivo Correlations Based on USP IV Dynamic Dissolution Applied to Four Sodium Montelukast Products

**DOI:** 10.3390/pharmaceutics13050690

**Published:** 2021-05-11

**Authors:** Mercedes Prieto-Escolar, Juan J. Torrado, Covadonga Álvarez, Alejandro Ruiz-Picazo, Marta Simón-Vázquez, Carlos Govantes, Jesús Frias, Alfredo García-Arieta, Isabel Gonzalez-Alvarez, Marival Bermejo

**Affiliations:** 1Farmacia y Tecnología Farmacéutica, Facultad de Farmacia, Universidad Complutense de Madrid, 28040 Madrid, Spain; mprietoescolar@hotmail.com (M.P.-E.); torrado1@ucm.es (J.J.T.); covadong@ucm.es (C.Á.); 2Departamento Ingeniería of Área Farmacia, Universidad Miguel Hernández de Elche, 03550 Alicante, Spain; alejandroruizpicazo@gmail.com (A.R.-P.); mbermejo@umh.es (M.B.); 3Laboratorios Normon SA, Tres Cantos, 28760 Madrid, Spain; msimon@normon.com (M.S.-V.); cgovantes@normon.com (C.G.); 4Clinical Pharmacology Service, Hospital Universitario la Paz, School of Medicine, Universidad Autónoma de Madrid, 28049 Madrid, Spain; jesus.frias@uam.es; 5Agencia Española de Medicamentos y Productos Sanitarios, 28022 Madrid, Spain; agarciaa@aemps.es

**Keywords:** montelukast, in vitro-in vivo correlation, flow-through cell

## Abstract

Montelukast is a weak acid drug characterized by its low solubility in the range of pH 1.2 to 4.5, which may lead to dissolution-limited absorption. The aim of this paper is to develop an in vivo predictive dissolution method for montelukast and to check its performance by establishing a level-A in vitro-in vivo correlation (IVIVC). During the development of a generic film-coated tablet formulation, two clinical trials were done with three different experimental formulations to achieve a similar formulation to the reference one. A dissolution test procedure with a flow-through cell (USP IV) was used to predict the in vivo absorption behavior. The method proposed is based on a flow rate of 5 mL/min and changes of pH mediums from 1.2 to 4.5 and then to 6.8 with standard pharmacopoeia buffers. In order to improve the dissolution of montelukast, sodium dodecyl sulfate was added to the 4.5 and 6.8 pH mediums. Dissolution profiles in from the new method were used to develop a level-A IVIVC. One-step level-A IVIVC was developed from dissolution profiles and fractions absorbed obtained by the Loo–Riegelman method. Time scaling with Levy’s plot was necessary to achieve a linear IVIVC. One-step differential equation-based IVIVC was also developed with a time-scaling function. The developed method showed similar results to a previously proposed biopredictive method for montelukast, and the added value showed the ability to discriminate among different release rates in vitro, matching the in vivo clinical bioequivalence results.

## 1. Introduction

Montelukast is a weak acid drug characterized by its low solubility in the range of pH 1.2 to 4.5, which may lead to dissolution-limited absorption. Due to its pH-dependent solubility (from 0.18 µg/mL at pH 1.2 to 0.24 mg/mL at pH 7.5 [1]), in vitro conventional dissolution tests may be not useful to predict the in vivo behavior of montelukast products, as they do not reflect the physiological pH changes during intestinal transit or the actual luminal fluid volumes. Dynamic dissolution methods, such as a USP IV flow through apparatus, allows the pH changes in the intestinal lumen and the hydrodynamic conditions and volumes of the gastrointestinal track to be mimicked. In this work, plasma concentrations of montelukast from two different bioequivalence trials were used to develop a level-A IVIVC. In a first trial, although conventional dissolution results with the rotating paddle method suggested product similarity, the Test 1 formulation failed to show bioequivalence. Therefore, a second trial was performed with two other test formulations (Test 2 and Test 3), where Test 2 proved to be bioequivalent. Due to the lack of in vitro-in vivo correlation (IVIVC) with the conventional rotating paddle method, a new dissolution test method was designed.

The conventional dissolution test of montelukast is based on additions of surfactants and neutral pH. Moreover, a flow-through cell dissolution test was recently proposed as a suitable method to study in vitro drug release from montelukast tablet formulation [1]. Although the method proposed by Okumu et al. [1] was useful to establish an in vivo predictive mathematical model, this method was based on complex mediums with bile salts and lecithin. Moreover, the validity of the method proposed by Okumu et al. [1] was tested in GastroPlus™ and only one formulation (i.e., dissolution rate) was linked with the in vivo outcome. The aim of this work was to develop a new dissolution test for montelukast film-coated tablets with simple surfactants. In addition, the level-A correlation was developed in two steps using Excel spreadsheets and the DDsolver tool [2], and in one step with a custom-made differential equation model implemented in Phoenix WinNonlin V8 (Certara USA, Princeton, NJ, USA). For the dissolution medium, only conventional buffers salts and sodium dodecyl sulphate were used. Once established, the IVIVC level-A correlation can be very useful in manufacturing quality-control processes, in new formulation development, or to guide formulation changes [3,4].

## 2. Materials and Methods

### 2.1. Chemicals

If not specified, they are of analytical grade.

### 2.2. Tested Formulations

Four different film tablet formulations of sodium montelukast equivalent to 10 mg of montelukast were studied. The reference formulation (Ref) was Singulair^®^ batch 283,358 from Merck Sharp & Dohme España and compared with three different test tablet formulations. Test tablet formulations were prepared by the Spanish Normon pharmaceutical company, described in this paper as Test 1, Test 2, and Test 3.

### 2.3. In Vitro Studies

Dissolution studies were performed using a rotating paddle apparatus method 2 from European Pharmacopoeia 6th ed. (Erweka DT80, Heusemstamm, Germany) and a flow-through large cell method 4 from European Pharmacopoeia (Sotax, Aesch, Switzerland). Dissolution studies in the rotating paddle equipment were conducted at a rotational speed of 50 rpm using 900 mL of various dissolution media at pH 1.2, 4.5, and 6.8. Buffer compositions were as follows:-A 1.2 pH buffer was obtained with a final concentration of 0.1 M HCl and 0.05 M NaCl;-Two different acetate 4.5 pH buffers were prepared at sodium dodecyl sulphate (SDS, 85%, Eur. Ph., Panreac, Barcelona, Spain) concentrations of 0.2 and 1.5% *w*/*v*. Acetate buffer was prepared with a final composition of sodium acetate 0.022 M and acetic acid 0.25 M. NaOH was added to obtain 4.5 pH; and-Tris 6.8 buffer with 0.2% SDS was used as dissolution medium. Initially conventional potassium phosphate buffer was prepared, but due to precipitation of SDS and the consequent decrease in dissolution of the montelukast, the phosphate buffer was replaced by tris buffer. Tris buffer was prepared containing tris(hidroximetil)amino methane 0.052 M and sodium anhydrous acetate 0.06 M. Sulfuric acid was added to adjust the pH.

Dissolution studies in the flow-through cell were done at a mean flow of 5 mL/min in an open design with the following dissolution media:From 0 to 15 min 1.2 pH medium;From 15 to 60 min acetate 4.5 pH with 1.5% *w/v* SDS; andFrom 60 to 210 min at 6.8 pH acetate buffer with 0.2% *w/v* SDS.

SDS was chosen to avoid the difficulties associated with mixtures of bile salt–lecithins and due to being an anionic surfactant as bile acids.

All dissolution studies were performed at 37 + 0.5 °C. Withdrawn samples were filtered and after appropriate dilution assayed for montelukast UV spectrophotometrically (Beckman DU-6, Brea, CA, USA) at 255.5 and 225 nm for samples of pH 4.5 and 6.8, respectively. The analytical method was validated and each dissolution profile was taken from 6 units.

Comparison between in vitro dissolution results of test formulations and the reference one was done by the similarity factor f2 [5].

### 2.4. In Vivo Studies

Two different clinical bioequivalence studies were performed in the Phase I Clinical Trial Unit of the La Paz Hospital, School of Medicine, Autonomous University of Madrid, Spain, following the updated Declaration of Helsinki, with the approval of the Ethical Committee for Clinical Research and the Spanish Agency for Medicines and Health Care Products. EudraCT 2008-004004-32 (study ethic approval code N-MON-08-136, 3 July 2008); EudraCT 2009-013451-30 (Study etic approval code N-MON-09-148, 2 July 2009). Both trials were designed as randomized, open label, and single dose. The first one was done with 36 volunteers and the second one with 24 volunteers. All volunteers were non-smoker healthy males or females. The subjects were determined to be in good health by physical examination, with a complete blood count, urinalysis, and serum test on hepatic and renal function. The age, height, body weight, and body mass index were 23.1 years, 172.2 cm, 74.5 kg, and 24.1 kg/m^2^, respectively, for the first trial, and 24.5 years, 169.2 cm, 65.6 kg, and 22.6 kg/m^2^, respectively, for the second trial. The volunteers were asked to abstain from taking any drug, including OTC products, for at least 1 week prior to or during the study. Written informed consent was obtained from subjects after explaining the nature and purpose of the study. After fasting overnight for 10 h, a dose of 10 mg of montelukast was orally administered with 200 mL of tap water. In the first bioequivalence study, Test 1 and the reference formulations were studied. In the second clinical study the test formulations numbers Test 2 and Test 3 were compared with the reference formulation. Blood was drawn before dosing and at 0.5, 1, 1.5, 2, 2.5, 3, 3.25, 3.5, 3.75, 4, 4.25, 4.5, 5, 6, 8, 12, 16, 20, and 24 h after dosing through an in-dwelling catheter placed in an antecubital vein in the forearm. The blood samples were centrifuged and the plasma was collected and stored at −20 °C until assayed. Volunteers were allowed to take water ad libitum. The first meal was served 5 h after dosing. Beverages and food containing caffeine were not permitted during the entire course of the study. The wash-out period was 7 days. Plasmatic levels of montelukast were assayed by a validated fluorescence chromatography procedure by the Research and Development department of Laboratorios Normon S.A. (Madrid, Spain). Data from the first bioequivalence study was normalized in relation to the data from the second one with the common reference formulation Singulair^®^ according to the IVIVC guide [6].

### 2.5. Pharmacokinetic Analysis

The individual elimination rate constant (K) was calculated from the slope of the log-linear phase of the plasma concentration–time curve using linear regression. The area under the plasma concentration–time curve from 0 to 24 h (AUC_0–24_) was calculated using the trapezoidal rule. The area under the plasma concentration–time curve from 24 h to ∞ (AUC_24–∞_) was calculated as C_24_ divided by K. C_max_ and T_max_ were the observed values. C_max_ and AUC ratios and 90% confidence intervals are reported in Table 1.

As plasma levels of montelukast products were obtained in two different BE studies, it was necessary to normalize them to account for the inter-subject variability in pharmacokinetic parameters across studies. Reference product data were used for that purpose under the assumption of the bioequivalence of both reference batches. A second issue was the different sampling schemes in both clinical trials. As sampling intervals were short, a combined set of sampling times was generated from both studies; thus, the missing plasma concentrations were estimated by linear interpolation between two consecutives sampling times. Reference product concentration ratios at each time point (Ref2/Ref1) were used as factor to correct the plasma levels of study 1 (Cp_corrected = Cp_observed*Ref2/Ref1).

Disposition parameters of montelukast were estimated from IV data (7 mg bolus) from Zhao et al. [7] in Phoenix Winnonlin, fitting a two-compartment open model. The estimated parameters are summarized in Table 2.

These parameters were used to perform a Loo–Riegelman analysis to estimate fractions absorbed (bioavailable fractions) [8].

Fractions were estimated using the highest estimated AUC from zero to infinity across formulations, to account for the different relative bioavailability across products.

Analysis was performed over the average plasma profiles, as the results do not change dramatically when differences in lag times or tmax (time to maximum plasma concentration) across subjects are small [9].

### 2.6. Dissolution Profile Modeling

A three-parameter Weibull model was fitted to the experimental data.
(1)$$F_{diss}=F_{max}*\left(1-e^{\left(-\left(\frac{t^{\beta }}{\alpha }\right)\right)}\right)$$

To account for the slight difference in time scale of the in vitro and in vivo dissolution process, A Levy plot was constructed. A Levy plot represents the relationship between the in vitro and in vivo times needed for dissolution/absorption of a given fraction. Times for in vitro dissolution of a given fraction (*f_abs_*) were estimated with the Weibull equation.
(2)$$t_{vitro}=\sqrt[{\beta }]{\alpha *\left(-1\right)*ln\frac{F_{max}-f_{abs}}{F_{max}}}$$

The plot was constructed using data up to 3.5 h, as at later points correlation was not a single function. In vitro times were transformed into their equivalent in vivo times and the Weibull function was fitted again to the dissolved fractions at the corrected times to get the corrected Weibull parameters.

### 2.7. Two-Step IVIVC Model Development

The Weibull time-corrected parameters were used to estimate fractions dissolved at the in vivo sampling times in order to construct the level-A in vitro-in vivo correlation, i.e., the relationship between the fractions dissolved and absorbed at the same times.

Level-A IVIVC was constructed with dissolved and absorbed fractions up to 3.5 h: Once the level-A correlation was obtained, predicted fractions absorbed from fractions dissolved and the level-A equation was convoluted again using the inverse Loo–Riegelman procedure [10] to get the predicted plasma levels.

The IVIVC level-A correlation was statistically studied in Excel (Microsoft Office 2007, USA) [11,12]. The IVIVC was studied either with the whole data set (four products) as an exploratory study to ascertain whether all of them behaved similarly and in a second phase with the data from three montelukast tablet formulations: reference, Test 1, and Test 3, to use Test 2 as an external validation set. U.S. Food and Drug Administration (FDA) criteria of predicted error (PE) were used to determine the level of IVIVC [6]. A PE value smaller than 10% was considered a correct model fit. The estimation of prediction error (PE) was done externally with the individual data of the Test 2 formulation, which was not used in the development of the IVIVC.

The percent prediction error (%PE) was calculated as
%PE = [(Observed value − Predicted value)/Observed value] × 100 (3)

### 2.8. One-Step IVIVC Model Development

For the one-step IVIVC model, the same disposition parameters as in the two-step approach were used. The one-step approach previously described by Buchwald was adapted and used [13]. The Weibull model fitted to the in vitro dissolution profiles was used to get the in vitro dissolution rates (*r_diss_*) that were linked to the in vivo input (dissolution) rates(*r_t_*) using a link function:(4)$$r_{t}=\varphi_{t}*sc*r_{diss\left(s0*t^{s1}\right)}$$
where *φ_t_* is the absorption cut-off.
(5)$$\varphi_{t}=\frac{e^{\left(-eta*\left(t-tcut\right)\right)}}{1+e^{\left(-eta*\left(t-tcut\right)\right)}}$$
where *eta* and *tcut* are the parameters of the cut-off function.

*SC* is the extent scaling factor, with parameters *SC*1 and *B*:*SC* = *SC*1 × *Fmax^B^*(6)
where *s*0 and *s*1 are the parameters of the time-scaling function:*tvitro* = *s*0 × *t*^*s*1^(7)
where *t* are in vivo times for a given fraction absorbed and *tvitro* is the in vitro times for the equivalent dissolved fraction.

*s*0, *s*1, *SC*1, *B*, *eta*, and *tcut* were estimated by curve fitting.

To link in vitro and in vivo dissolution rates, the deconvoluted fractions absorbed estimated by the Loo–Riegelman method were modeled using a Weibull function. In vitro and in vivo Weibull parameters were correlated with linear functions as follows:Parameter vivo’ = a + m × Parameter vitro(8)
(9)$$r_{diss}=\frac{\beta^{\prime }*F_{max}*t_{vitro}^{\left(\beta^{\prime }-1\right)}*e^{-\left(\frac{t_{vitro}^{\left(\beta^{\prime }-1\right)}}{\alpha^{\prime }}\right)}}{\alpha^{\prime }}$$

Differential equations describing the time evolution of amounts in the central and peripheral compartments were the following:(10)$$\frac{dQ_{c}}{dt}=dose*\varphi_{t}*sc*r_{diss\left(t_{vitro}\right)}-k10*Q_{c}-k12*Q_{c}+k21*Q_{p}$$
(11)$$\frac{dQ_{c}}{dt}=k12*Q_{c}-k21*Q_{p}$$
(12)$$C_{plasma}=\frac{Q_{c}}{V_{ss}}$$
where *Q_c_* and *Q_p_* are the amounts in the central and peripheral compartments and Cplasma is the observed plasma level. Plasma concentrations were estimated from the amounts by dividing by the steady-state distribution volume.

The model was fitted to the plasma levels of all formulations simultaneously to estimate the 6 parameters of the one-step IVIVC, namely, *eta*, *tcut*, *s*0, *s*1, *SC*1, and *B*.

Correlations were developed with 3 products (Reference, Test 1, and Test 3; Test 2 was excluded and used for external validation) and with the 4 products.

## 3. Results

### 3.1. In Vitro and In Vivo Data

Figure 1 shows the original and normalized plasma concentrations of montelukast formulations.

Figure 2 summarizes the Loo–Riegelman absorption analysis. The absorption profiles were clearly different for the non-bioequivalent formulation (Test 1 and Test 3), whereas the bioequivalent one (Test 2) showed a very similar absorption rate with a slightly higher asymptotic value in concordance with the Cmax and AUC ratios.

Figure 3, Figure 4 and Figure 5 show the dissolution results obtained with the rotational paddle dissolution equipment. Figure 3 and Figure 4 show the effect of SDS concentration at 4.5 pH and Figure 5 shows the effect of pH 6.8.

As can be seen in Figure 3, Figure 4 and Figure 5, it was difficult to classify the tested tablet formulations related to their release rate. Although usually Test 3 formulation is the slower one, it was not easy to establish the relationship among the other three formulations. The similarity factor f2 was calculated for the three formulations and the four different dissolution conditions studied and the results are reported in Table 3.

Figure 6 shows the dissolution profiles of all formulations in the flow-through apparatus with pH transition. The dissolution profiles showed a rank order parallel to the results obtained in the in vivo clinical evaluation.

### 3.2. Two-Step IVIVC

To account for the slight difference in time scale of in vitro and in vivo dissolution, a Levy plot was constructed (see Figure 7). A Levy plot represents the relationship between the in vitro and in vivo times needed for the dissolution/absorption of a given fraction. The times for in vitro dissolution of a given fraction (*f_abs_*) were estimated from the Weibull equation (Equation (2)).

The plot was constructed using data up to 3.5 h, as at later points the correlation was not a single function. In vitro times were transformed into their equivalent in vivo times and the Weibull function was fitted again to the dissolved fractions at the corrected times.

The time-corrected Weibull parameters are summarized in Table 4.

The parameters in Table 4 were used to estimate fractions dissolved at the in vivo sampling times in order to construct the level-A in vitro in vivo correlation, i.e., the relationship between the fractions dissolved and absorbed at the same time.

Level-A IVIVC was constructed with dissolved and absorbed fractions up to 3.5 h. The two-step IVIVC calculated with the four products is represented in Figure 8. Once the level-A correlation was obtained, the predicted fractions absorbed from the fractions dissolved and the level-A equation were convoluted again using the inverse Loo–Riegelman procedure [10] to get the predicted plasma levels. The predicted and experimental plasma levels are represented in Figure 9 with the internal prediction errors.

Correlation was also developed using Reference, Test 1, and Test 3, whereas Test 2 was excluded and used later for external validation. The steps were the same as before. The Levy plot without Test 2 formulation is depicted in Figure 10. The Two-step level-A IVIVC without Test 2 formulation is depicted in Figure 11. The predicted plasma levels with the IVIVC obtained with 3 formulations and using Test 2 as external validation is represented in Figure 12.

With t vitro scaled to t vivo, the Weibull function was fitted again to the scaled dissolution profiles. The scaled parameters are shown in Table 5. These scaled parameters were used to estimate the fractions dissolved at the in vivo sampling times to obtain the IVIVC represented in Figure 11.

### 3.3. One-Step IVIVC

As in the two-step approach, a Weibull three-parameter model was fitted to the fractions dissolved and fractions absorbed. The in vitro and in vivo Weibull parameters were correlated with linear functions, whose parameters are summarized in Table 6.

The predicted in vivo Weibull parameters were used in Equation (7) to fit Equations (8)–(10) to the experimental plasma levels in order to get the parameters of the IVIVC relationship.

One-step correlation was developed with three formulations—Reference, Test 1, and Test 3—in order to use Test 2 for external validation, and one-step IVIVC was also developed with all the formulations.

Figure 13 and Figure 14 summarize the experimental and model predicted plasma levels with three and four products, respectively, and the model parameters are shown in Table 7 and Table 8.

Finally, in order to compare the performance of the developed IVIVC to the PBPK approach used by Okumu et al. [1], the reference in vitro dissolution profile was used as an input function in our mathematical model and the parameters and the predicted plasma levels were compared to their in vivo clinical data. The results are presented in Figure 15.

## 4. Discussion

The present study proposes a dynamic dissolution method based on USP IV apparatus to test montelukast formulations as a dissolution method, and media based on a USP 2 apparatus did not show in vivo relevance. A single pH condition is not adequate to predict the in vivo behavior of an ionizable drug with a pH-dependent dissolution rate. This fact was observed for other poorly soluble acids and bases [14,15].

According to the initial in vitro dissolution results obtained with the different conditions tested with method 2, the Test 1 formulation was initially selected for a first in vivo bioequivalence study. In this first trial, Test 1 was found to be supra-available, and a second in vivo trial with the two other test formulations was performed in which Test 2 formulation proved to be bioequivalent.

The comparison data of the f2 in vitro results reported in Table 3 with the in vivo data shown in Figure 5 indicate that the flow-through method has a better in vivo predictability than the rotating paddle method.

Despite the technical difficulties associated with the use of this high concentration of SDS that have been reported [16], we succeeded in the use of this media as reported by other authors [17].

In order to further explore this method’s utility, a level-A IVIVC development was planned.

To do so, the present study dealt with the combination of data obtained in two different BE studies. To allow the adequate comparison of both data sets, a normalization procedure based on the reference’s concentration ratios was used [18]. After normalization, both reference profiles were superimposable and the difference between test products was on the same extent scale.

As can be observed by comparison of the Loo–Riegelman fraction absorbed profiles in Figure 2 and the dissolved fractions in Figure 6, the complete in vivo dissolution/absorption for the fastest formulation (Test 1) needed 4 h in vivo, whereas in vitro was almost completed in 3 h. The difference was not very high but it made the time-scaling approach with a Levy plot necessary. Either in the one-step or the two-step approach, prediction errors were beyond the acceptable limits (>15%) when no time-scaling was used.

To construct a Levy plot, it is necessary to model dissolution profiles with an adequate equation in order to estimate dissolved fractions at any desired time. This approach was first proposed by Gonzalez-Garcia et al. [19] and described in detail by Cardot et al. as the Inverse Release Function approach [18]. Without the time-scaling factor obtained thanks to the Levy plot (Figure 7 and Figure 10), the fraction absorbed versus time profile and the fractions dissolved versus time one were not superimposable and in consequence it was not possible to obtain a good two-step linear level-A IVIVC. After the time scaling, good linear IVIVC correlations were obtained either with 3 or 4 formulations (Figure 8 and Figure 11).

In a previous work, Okumu et al. [1] developed a biopredictive method based on apparatus IV and pH transition with biorelevant media for montelukast dissolution. The authors used a PBPK model approach in Gastroplus to predict plasma levels from a clinical study. A limitation of that study is that only a reference formulation was used; thus, the ability of the method to discriminate among formulations with different release rates was not demonstrated. The rationale for the method development was based on the pH-dependent solubility of montelukast, and the inclusion of natural surfactants improved the in vivo predictions due to the lipophilicity of the drug, which in combination with the pH determines dissolution rate changes as the product transits in the gastrointestinal system. In this work, the aim was to test a similar method but avoid the use of bile salts and lecithins based on the hypothesis that a synthetic surfactant could be used to mimic the solubilization effect of bile salts and lecithin micelles.

To compare the performance of both methods, the dissolution profile of the reference formulation (Singulair^®^) in Okumu’s method and the proposed one, both dissolution profiles are represented in Figure 6. As can be observed, the reference profiles were similar in extent and slope up to 2 h and the difference in the asymptotic value, higher with Okumu’s method, could be accounted in their final transition to pH 7.5 (at which drug solubility is higher), which was not used in our method.

Another difference from the previously published approach was the mathematical procedure for the IVIVC characterization. In this paper, we used the two-step method based on deconvolution and a one-step procedure based on a semi-empirical differential equation model. The pharmacokinetic model is a modification of the one proposed by Buchwald [13]. Buchwald proposed the use of an input function based on the derivative of the Weibull dissolution model, a cut-off function to account for the transit out of the absorption window and time-scale and extent scaling factors. All these features were used in our model with the difference of a non-linear time function scale. In the case of montelukast, the cut-off absorption function accounted for the lower absorption of the drug in the distal portions of the gastrointestinal system. Actually, the cut-off time was about 12 h, which corresponds to the transit on the distal parts of the large intestine, where the lower available volume and permeation surface may decrease the montelukast dissolution/absorption rate. The extent scale factor, SC1, is an empirical representation of the first pass effect, described in Okumu’s model.

To further explore the similarity of the proposed in vitro method with Okumu’s in combination with their PBPK and our semi-empirical model, the present IVIVC was used to predict montelukast plasma levels from the original dissolution profile obtained with Okumu’s method, and the predicted levels were compared with the clinical data used by them.

As can be seen in Figure 15, despite our mathematical model having been developed with a different set of formulations and in different dissolution conditions, the predictions closely matched the in vivo profile, indicating montelukast absorption is dissolution limited and our method produces a similar outcome with synthetic surfactant.

Regarding the predictability of the one-step approach versus the two-step method, the prediction errors were on average slightly lower in the differential equations-based method, in accordance with our previously published comparisons [20], in which the difficulty of getting a good linear level-A IVIVC with the two-step approach was demonstrated when the relationship between the in vitro and in vivo dissolution rate is non-linear.

## 5. Conclusions

The developed method showed similar results to a previously proposed biopredictive method for montelukast, and as added value showed the ability to discriminate among different release rates in vitro, matching the in vivo clinical bioequivalence results.

## Figures and Tables

**Figure 1 pharmaceutics-13-00690-f001:**
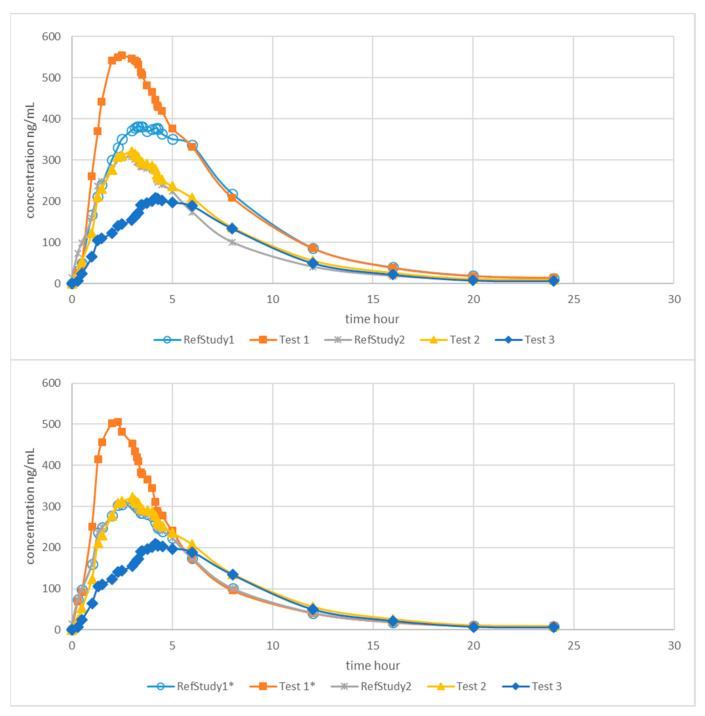
Original (**top panel**) and * normalized (**bottom panel**) montelukast plasma concentrations.

**Figure 2 pharmaceutics-13-00690-f002:**
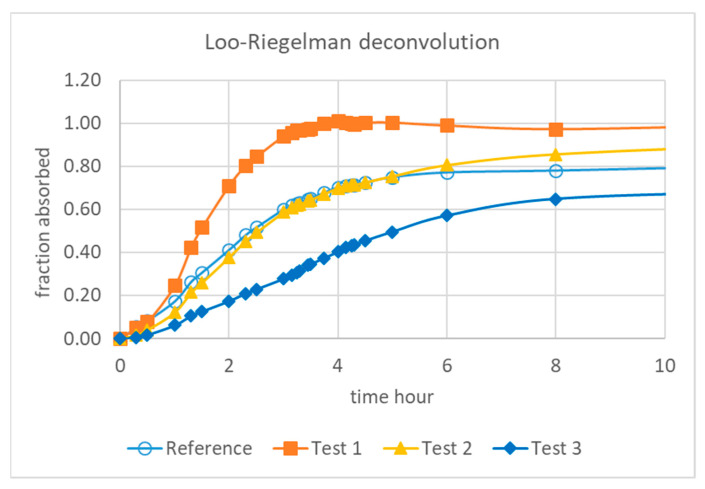
Fractions absorbed estimated with the Loo–Riegelman method.

**Figure 3 pharmaceutics-13-00690-f003:**
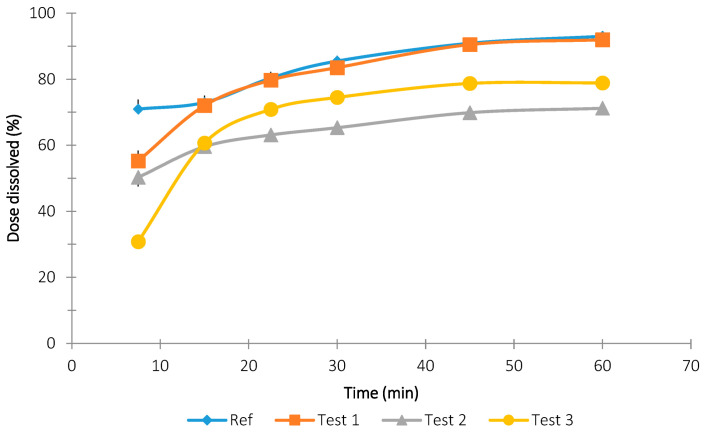
Mean dissolution results and standard deviation obtained with the rotational paddle dissolution equipment at 4.5 pH with 1.5% *w*/*v* SDS.

**Figure 4 pharmaceutics-13-00690-f004:**
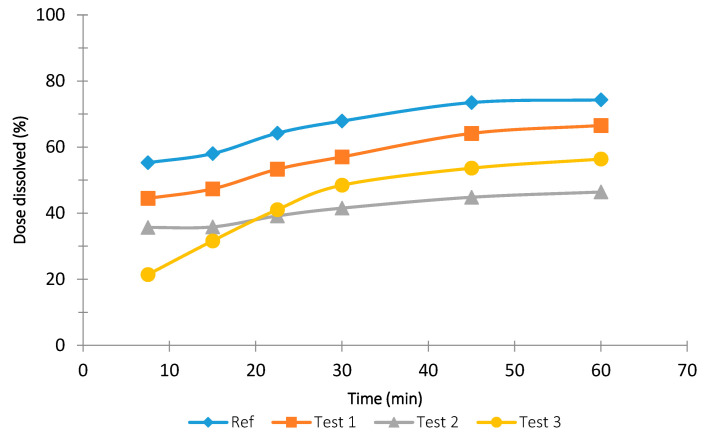
Mean dissolution results and standard deviation obtained with the rotational paddle dissolution equipment at 4.5 pH with 0.2% *w*/*v* SDS.

**Figure 5 pharmaceutics-13-00690-f005:**
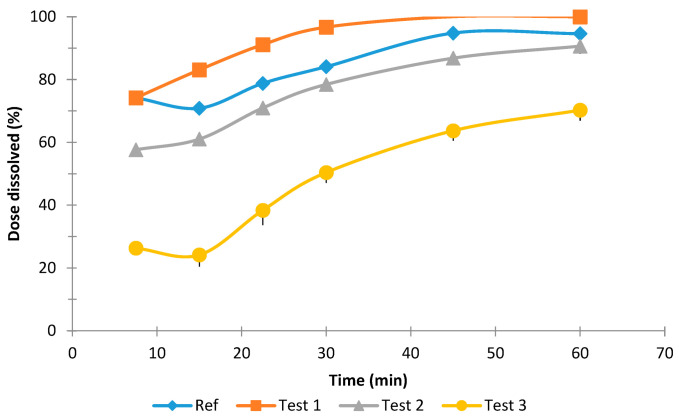
Mean dissolution results obtained with the rotational paddle dissolution equipment at 6.8 pH and 0.2% *w*/*v* SDS.

**Figure 6 pharmaceutics-13-00690-f006:**
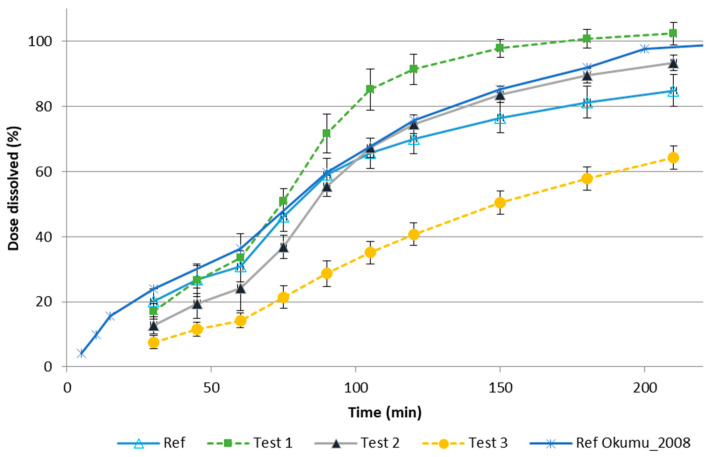
Dissolution profiles obtained in the flow-through apparatus with the pH transition and SDS of the reference product and the generic formulations. For comparison, the dissolution profile of the reference product in the method proposed by Okumu et al. is [1] overlapped.

**Figure 7 pharmaceutics-13-00690-f007:**
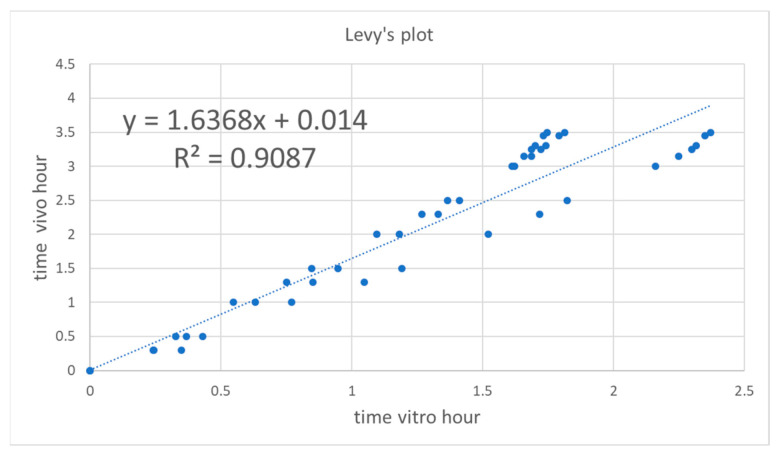
Levy’s plot for Reference, Test 1, Test 2, and Test 3.

**Figure 8 pharmaceutics-13-00690-f008:**
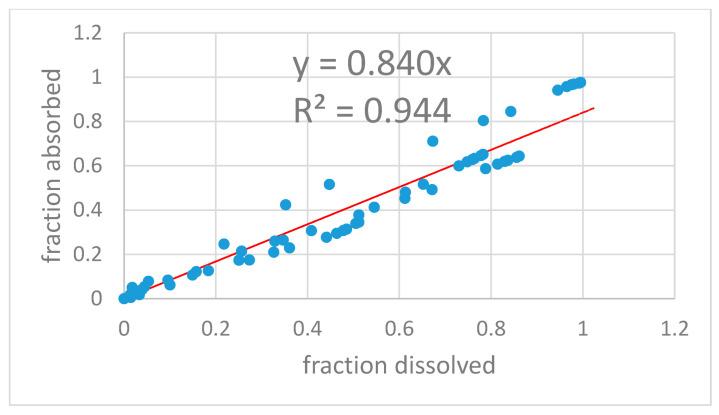
Two-step level-A two-step IVIVC for the four montelukast products.

**Figure 9 pharmaceutics-13-00690-f009:**
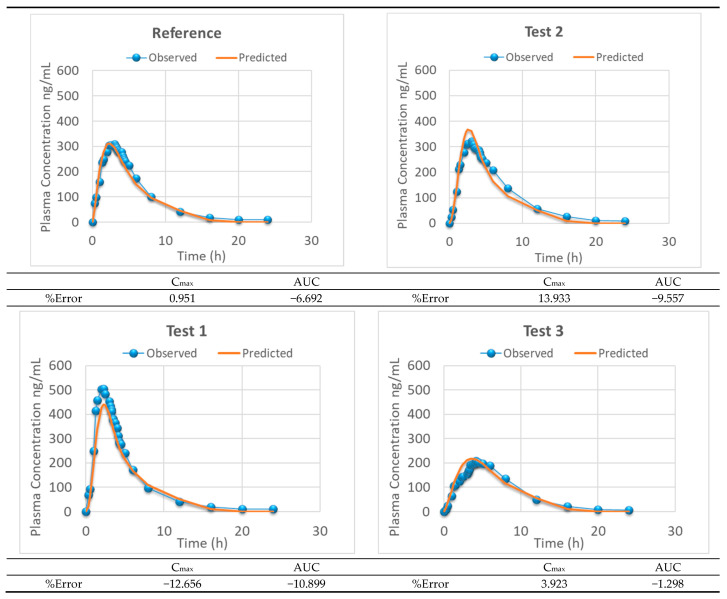
Experimental versus IVIVC-predicted plasma levels representing the two-step level-A IVIVC of the four assayed products and the internal prediction errors.

**Figure 10 pharmaceutics-13-00690-f010:**
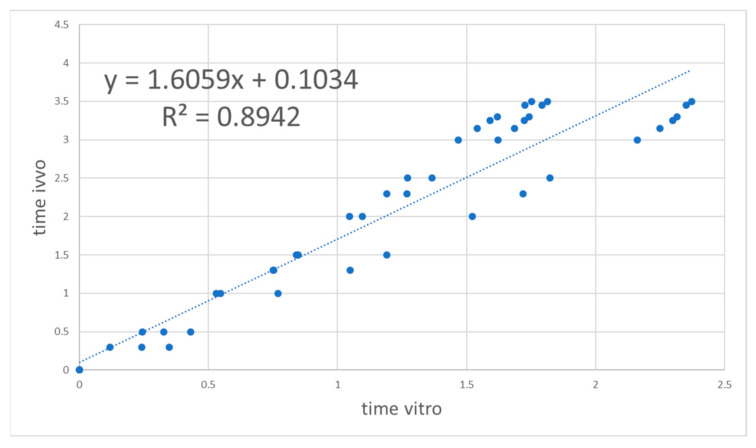
Levy plot constructed with Reference, Test 1, and Test 3 products.

**Figure 11 pharmaceutics-13-00690-f011:**
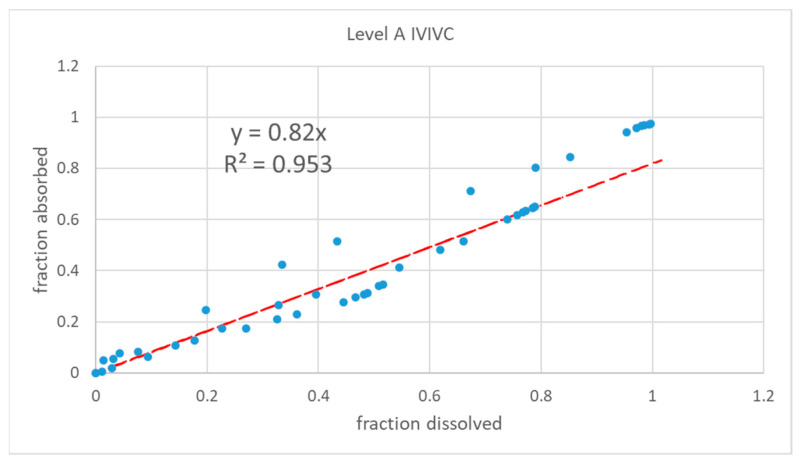
Two-step level-A IVIVC developed with Reference, Test 1, and Test 3 products.

**Figure 12 pharmaceutics-13-00690-f012:**
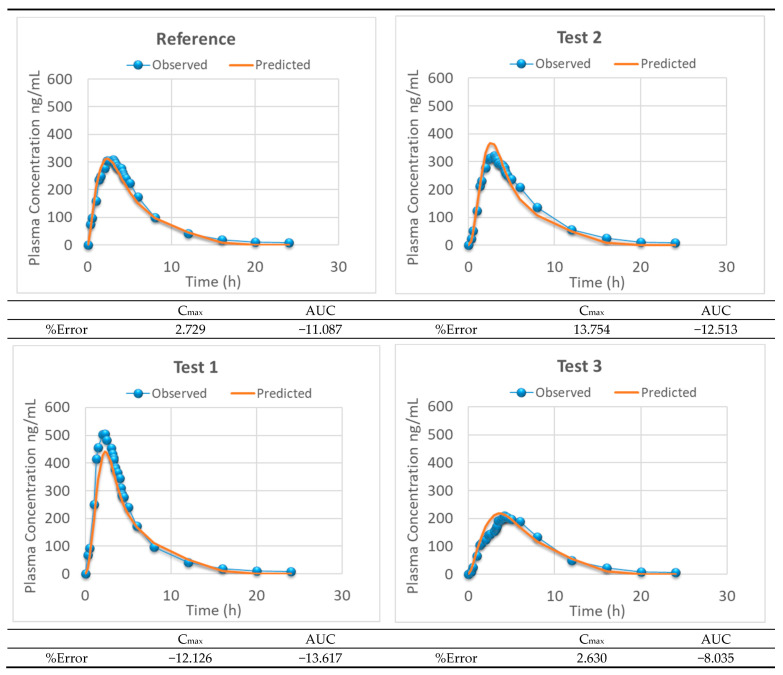
Experimental versus IVIVC-predicted plasma levels representing the two-step level-A IVIVC obtained with Reference, Test 1, and Test 3. Internal prediction errors and external prediction error for Test 2.

**Figure 13 pharmaceutics-13-00690-f013:**
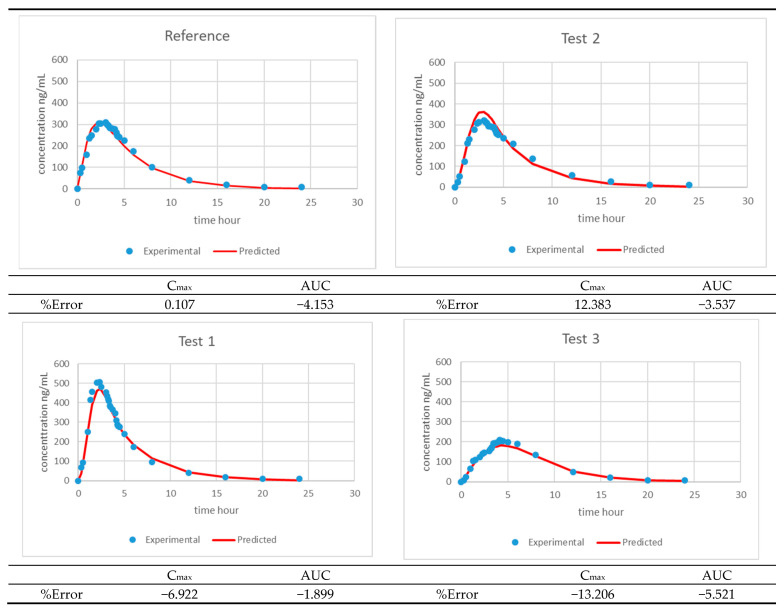
Predicted (solid line) and experimental (dots) montelukast plasma levels representing the one-step level-A IVIVC developed with 3 formulations. The AUC and Cmax internal prediction errors and the external prediction error for Test 2 are shown.

**Figure 14 pharmaceutics-13-00690-f014:**
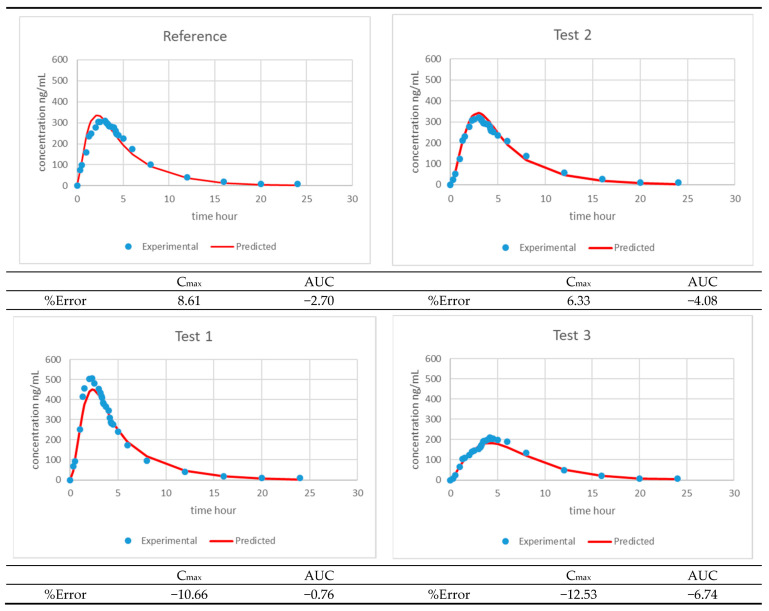
Predicted (solid line) and experimental (dots) montelukast plasma levels representing the one-step level-A IVIVC developed with 4 formulations.

**Figure 15 pharmaceutics-13-00690-f015:**
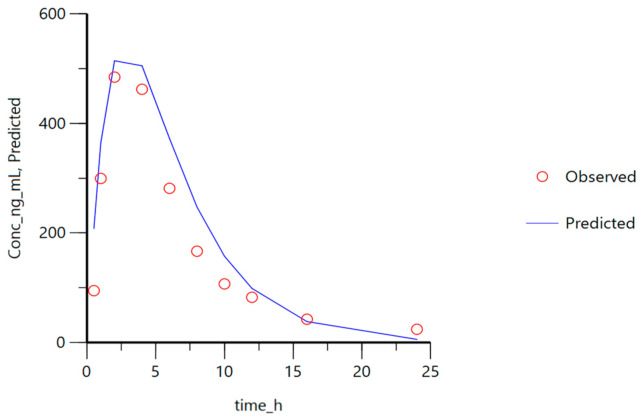
Plasma levels predicted with the proposed one-step IVIVC (based on four products) and the dissolution profile from the reference product from Okumu et al. [1]. The observed plasma levels correspond to the in vivo clinical data used by the same authors.

**Table 1 pharmaceutics-13-00690-t001:** Summary of the AUC and C_max_ ratios from both BE studies.

Product/Study		Ratio Test/Ref	90% CI Lower Limit	90% CI Higher Limit
Test 1/Study 1	Ln(C_max_)	131.67	118.84	145.89
	Ln(AUC_0–t_)	121.04	111.88	130.94
Test 2/Study 2	Ln(C_max_)	96.76	80.56	116.22
	Ln(AUC_0–t_)	96.90	81.47	115.25
Test 3/Study 2	Ln(C_max_)	68.99	57.44	82.87
	Ln(AUC_0–t_)	70.96	59.66	84.40

Table 1 BE 90% confidence intervals (CI) of Cmax and AUC ratios.

**Table 2 pharmaceutics-13-00690-t002:** Montelukast two-compartment pharmacokinetic parameters.

Parameter	Estimate	Std Error	CV%
Vc mL	2754.74	92.38	3.35
K10 h^−1^	0.65	0.04	6.03
K12 h^−1^	1.15	0.15	12.74
K21 h^−1^	0.90	0.13	14.32
Alpha h^−1^	2.46	0.27	10.84
Beta h^−1^	0.24	0.02	10.48
Vss h^−1^	6265.59	368.46	5.88

Vc: central compartment volume, Vss: steady state volume; K10: elimination rate constant from central compartment; K12: distribution rate constant from central to peripheral compartment; K21: distribution rate constant from peripheral to central compartment; alpha and beta: fast and slow disposition rate constants, respectively. Std error: standard estimation error; CV%: coefficient of variation.

**Table 3 pharmaceutics-13-00690-t003:** Similarity factor f2 calculated for the three test formulations at the four different dissolution conditions with the two pharmacopoeia dissolution methods.

	Test 1	Test 2	Test 3
Rotational paddle at 4.5 pH and 0.2% SDS	49.6	29.9	30.9
Rotational paddle at 4.5 pH and 1.5% SDS	54.9	37.1	32.6
Rotational paddle at 6.8 pH and 0.2% SDS	49.9	49.3	19.6
Flow-through cell	45.6	58.9	31.2

**Table 4 pharmaceutics-13-00690-t004:** Time-corrected Weibull parameters.

Parameter	Reference	Test 1	Test 2	Test 3
α	2.916	4.180	5.395	7.000
β	1.545	2.163	2.108	1.656
Fmax	0.863	1.023	0.930	0.753

**Table 5 pharmaceutics-13-00690-t005:** Time-scaled Weibull parameters without Test 2 in the Levy plot.

Parameter	Reference	Test 1	Test 2	Test 3
α	3.182	4.604	5.797	7.167
β	1.731	2.322	2.223	1.757
Fmax	0.843	1.016	0.922	0.720

**Table 6 pharmaceutics-13-00690-t006:** Slope (m) and intercept (a) of the linear correlations between the in vivo and in vitro Weibull parameters obtained by curve fitting, with the fractions dissolved and fractions absorbed.

	4 Products: Reference, Test 1, Test 2, Test 3			3 Products: Reference, Test 1, Test 2, Test 3		
	a	m	R2	a	m	R2
Fmax	−0.239	1.206	0.999	−0.239	1.206	0.999
α	−2.328	2.919	0.948	−2.156	2.934	0.966
β	0.773	0.464	0.336	0.169	0.856	0.998

**Table 7 pharmaceutics-13-00690-t007:** Parameters (and their standard error and coefficient of variation) of the IVIVC developed with 3 formulations.

	Value	Std Error	CV%
s0	1.069	0.049	4.57
s1	0.967	0.046	4.73
SC1	1.55 × 10^−5^	6.62 × 10^−6^	42.64
ETAL	3.623	450.797	>100%
TCUTL	9.585	8.708	90.85
B	1.402	0.095	6.78

**Table 8 pharmaceutics-13-00690-t008:** Parameters (and their standard error and coefficient of variation) of the IVIVC developed with 4 formulations.

	Value	Std Error	CV%
s0	1.015	0.051	5.07
s1	1.046	0.047	4.52
SC1	1.56 × 10^−5^	7.55 × 10^−6^	48.22
ETAL	3.737	8729.724	>100%
TCUTL	13.541	331.958	>100%
B	1.415	0.107	7.55

## Data Availability

Not applicable.

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
