# Peer review of "One and Two-Step In Vitro-In Vivo Correlations Based on USP IV Dynamic Dissolution Applied to Four Sodium Montelukast Products"

_pharmaceutics, 2021, doi:10.3390/pharmaceutics13050690_

Round 1
Reviewer 1 Report
as attached

Author Response
Thanks for your suggestion! We have improved the article according to your commnets. Graphs have been modified too as you recommend.
Attach the pont by point and the article with changes in red.

Reviewer 2 Report
1) You describe the following in the text: In order to improve the dissolution of montelukast sodium dodecyl sulfate is added to the 4.5 and 6.8 pH mediums.
Why did you choose to use sodium dodecyl sulphate as a dissolution aid?
Please comment on why you did not choose another dissolution aid such as Tween.
2) You have chosen Time scaling with Levy's plot for your experiment.
What is a Levy's plot?
Please add a definition of Levy's plot and explain the significance of Levy's plot in your research.
3) The results are listed in the Discussion section; please transcribe the description of the results in the Discussion section into the Results section.If it is difficult to proofread (write) the story as a story, why don't you write it as Results & Discussion?
4) Line 345-347 states the following.
As it can be seen in Figures 3 to 5 it is difficult to classify the tested tablet formulations related to their release rate. is the slower one, it is not easy to establish the relationship among the other three formulations.
Please discuss the reasons and add to the text.
5) Please add a note about Funding at the end of the text.
@Respond to comments politely. If you do, it will be an interesting study for many pharmaceutical researchers.
Author Response
Thanks a lot for your suggestions!
we are including them in the manuscript
1) You describe the following in the text: In order to improve the dissolution of montelukast sodium dodecyl sulfate is added to the 4.5 and 6.8 pH mediums.
Why did you choose to use sodium dodecyl sulphate as a dissolution aid?
SDS was chosen to avoid the difficulties associated with mixtures of bile salt-lecithins and for the fact of being an anionic surfactant as bile acids.
That information has been added to the manuscript.
Please comment on why you did not choose another dissolution aid such as Tween.
We agree with the reviewer in the proposition of tween as a suitable alternative, but we wanted to use a surfactant with the same ionization properties as bile acids
2) You have chosen Time scaling with Levy's plot for your experiment.
What is a Levy's plot?
Please add a definition of Levy's plot and explain the significance of Levy's plot in your research.
We kindly acknowledge the reviewer suggestion for improving the manuscript,
Definition of levy’s plot was already included in section 3.2 Two step leval A ivivc
Levy’s plot represents the relationship between the in vitro and in vivo times needed for dissolution/absorption of a given fraction.
Levy’s plot is represented in Figure 7.
But to explain in detail the significance in our research, following the recommendation of the reviewer, the following explanation has been included in the discussion section:
Without the time-scaling factor obtained thanks to the Levy’s plot (Figure 7) the fraction absorbed versus time profile and the fractions dissolved versus time one are not superimposable and in consequence is not possible to obtain a good two step linear level A IVIVC
3) The results are listed in the Discussion section; please transcribe the description of the results in the Discussion section into the Results section.If it is difficult to proofread (write) the story as a story, why don't you write it as Results & Discussion?
The description of the results has been moved to result section.
4) Line 345-347 states the following.
As it can be seen in Figures 3 to 5 it is difficult to classify the tested tablet formulations related to their release rate. is the slower one, it is not easy to establish the relationship among the other three formulations.
Please discuss the reasons and add to the text.
The next sentence has been added to the discussion
A single pH condition is not adequate to predict the in vivo behaviour of an ionisable drug with pH-dependent dissolution rate. This fact has been observed for other poorly soluble acids and bases.
5) Please add a note about Funding at the end of the text.
Acknowledgement section was already included in the place indicated by the journal:
Acknowledgments: This work was partially supported by a grant of the U.C.M. to the research group 910939 and partially supported by Agencia Estatal de Investigación and the European Union, through FEDER (Fondo Europeo de Desarrollo Regional), grant number SAF2016-78756 (AEI/FEDER, EU).

Round 2
Reviewer 2 Report
Thank you for your polite and genuine response to my comments.
Let's keep up the good work as the same scientist in the development of formulation science.
Sincerely,